# Reproducibility Study of 'Exacerbating Algorithmic Bias through Fairness Attacks'

## Reproducibility Summary

**Scope of Reproducibility**

The goal of this paper is to assess the reproducibility of experiments and results in the paper 'Exacerbating Algorithmic Bias through Fairness Attacks' by Mehrabi et al. (2020), from which the following claims are evaluated:

− Claim 1: The anchoring attacks reduce the fairness of an ML model trained on the three data sets German Credit, COMPAS and Drug consumption.

− Claim 2: The influence attack reduces the fairness of an ML model trained on the three data sets German Credit, COMPAS and Drug consumption.

**Methodology**

We used the code the authors published alongside their paper as a resource to understand the methodology of their experiments, which was only briefly touched upon in the original paper. Our contribution is to extrapolate the original method using the provided code and to use this to recreate the experiments, successfully obtaining similar results as the paper and supporting their claims.

**Results**

Our results followed similar patterns as those of the authors, which backs up their claims regarding the attacks. However, our results did slightly deviate from their results, meaning the original paper has some reproducibility issues in the context of our experimental setup.

**What was easy and what was difficult**

It was difficult to understand the experiments from the paper. In our specific setting it was not possible to obtain similar results following only the methodology of their paper. Recreating the data sets required several assumptions. Reorganizing the code was a challenge in and of itself, owing to a lack of documentation within the original code.

**Communication with original authors**

We had no direct contact with the authors. However, other research teams working on reproducing the same work provided us with a digital environment file supplied to them by the authors.

# 1 Introduction

Recent years have seen a rising interest in algorithmic fairness, which has led to different measures and definitions for characterizing fairness (Dwork et al., 2012; Hardt et al., 2016; Kusner et al., 2017; Verma and Rubin, 2018; Mehrabi et al., 2019). Areas in which algorithmic fairness has become prevalent include predicting whether prisoners are likely to re-offend upon release (Duwe and Kim, 2017) or whether an individual is likely to default on a loan payment (Ereiz, 2019).

In 'Exacerbating Algorithmic Bias through Fairness Attacks' by Mehrabi et al. (2020) it is claimed that machine learning (ML) models are not only susceptible to various malicious adversarial attacks targeting their accuracy, but also to those targeting the fairness of ML models. Mehrabi argues that a model's fairness is as important as its accuracy and research into adversarial attacks specifically designed to attack fairness is therefore warranted. To test the robustness of fairness methods intended to increase the fairness of an ML model, the researchers propose two novel data poising attacks on fairness, those being the anchoring attack and the influence attack.

The anchoring attack has two variations; random and non-random. The core concept is to place poisoned points near real data points of a data set, to skew the decision boundary of an ML model. These poisoned points are identical to the point they are placed close to, but with the opposite target label. The influence attack on fairness (IAF) aims to lower the fairness of an ML model by introducing fairness loss to the loss function. Maximizing for this loss function maximizes the covariance between the distance to the decision boundary and the sensitive features.

This paper investigates the reproducibility of the research of Mehrabi et al. (2020). Additionally, their claims regarding the two proposed fairness attacks the fairness of a targeted ML model will be tested, analyzed, and evaluated.

# 2 Scope of reproducibility

The main contribution of Mehrabi et al. (2020) is presenting two novel fairness attacks, called (random and non-random) anchoring attacks and influence attacks, and showing that these attacks more negatively impact the fairness scores of ML models than adversarial attacks on accuracy. To reproduce to work of the the original paper, the code and altered versions of three data sets, German Credit, Drug Consumption and COMPAS data sets accompanying the paper, which is publicly available on GitHub[1], are utilized. Fairness is quantified using the metrics statistical parity difference (SPD) (Dwork et al., 2012) and equality of opportunity difference (EOD) (Hardt et al., 2016), following the approach of Mehrabi et al. (2020).

The following are the main claims made within the original paper by Mehrabi et al. (2020):

− Claim 1: The anchoring attacks reduce the fairness of an ML model trained on the three data sets German Credit, COMPAS and Drug consumption.

− Claim 2: The influence attack reduces the fairness of an ML model trained on the three data sets German Credit, COMPAS and Drug consumption.

− Claim 3: Poisoning attacks designed to attack the accuracy of an ML model are not suitable as a fairness attack.

Claim 3 will not be considered in this paper, as the original authors mention it only briefly. They only evaluated whether influence attacks on accuracy had any effect on a model's fairness, without evaluating any other form of accuracy attack. In order to obtain results that can reject or support this claim, one would have to consider other adversarial attacks on accuracy, which is beyond the scope of this paper.

To demonstrate the effectiveness of their fairness attacks, the authors compare it to a fairness attack inspired by Solans et al. (2020). However, to thoroughly evaluate the effectiveness of the novel attacks, one would have to compare against multiple other concurrent works on adversarial attacks on fairness. Since we were only allocated four weeks for this project, this is also beyond the scope of this paper.

The focus of this paper will thus solely be on reproducing the novel attacks introduced by Mehrabi et al. and evaluating claims 1 and 2.

---

[1]https://github.com/Ninarehm/attack

# 3 Methodology

The authors' code, provided alongside the paper, includes a clear entry point as well as the data sets used for the discussed experiments. However, there were several issues with reproducing the experiments, such as a reliance on outdated Python libraries of which the new versions are not backwards-compatible. This is likely a result of the code being a combination of the code of previous papers that Mehrabi et al. (2020) based their research on, which resulted in a lack of documentation. Furthermore, information about data pre-processing is missing from the original paper, causing reproducibility issues. Only the attributes and the classification goal for each data set were clearly reported. Additionally, the number of features we discovered in the data sets provided by the authors did not match the number of features described in their paper. These issues required us to make multiple assumptions as we aimed to recreate these modified data sets from the original raw versions. The exact nature of these assumptions is further detailed in section 3.2. A list of all made assumptions is found in the Appendix. As a result of this obscurity regarding both the original method and the number of assumptions necessary to reconstruct the method, we decided not to re-implement the code in its entirety, instead making adjustments and additions to the original code to reproduce the original implementation. This is discussed in the next section.

To increase the scalability and maintainability of the code base, the intent was to employ the PyTorch framework instead of the TensorFlow framework used by the original authors. However, there were no straighforward substitutions for some TensorFlow functions, such as tf.truncated_normal_initializer and tf.variable. This would necessitate a change to some of the code's fundamental structures. As our approach is centered around utilizing the code provided by the authors, which, due to its complexity, required a significant amount of time to understand, there was a limited amount of time available for making such substantial modifications to the code.

## 3.1 Model descriptions

The model that the authors used to minimize the classification loss was not specified in the original paper. The authors' code, however, revealed that SciPy's fmin optimizer[2] was utilized as a minimizer for the experiments, which minimizes the loss by applying the Nelder-Mead algorithm (Nelder and Mead, 1965).

A **data poisoning attack** (DPA) has the goal of creating poisoned data set $D_p$ using the original clean data set $D_c$, such that the defender's test loss function $L(\hat{\theta}; D_{test})$ is maximized. To do so, iterative gradient steps are taken on each of the features of the poisoned data points $D_p$. The poisoned points are then projected to the feasible set $\mathcal{F}_\beta$ to avoid being detected by the defender's anomaly detector. According to the paper, as well as the algorithms in Figure 11, the feasible set is obtained by applying anomaly detector B; $F_b \leftarrow B(D_c \cup D_p)$. However, the anomaly detector B is not described in detail. Observing the code led to the assumption that the feasible set is determined by simply projecting the data onto a slab in close proximity to the target, shielding the attacker from anomaly detection.

This is not the first time that such gradient-oriented poisoning of data was implemented, as it was first explored using SVMs (Biggio et al., 2013), and in the following years extended to linear and logistic regression (Mei and Zhu, 2015b), topic modeling (Mei and Zhu, 2015a), collaborative filtering (Li et al., 2016), and neural networks (Koh and Liang, 2017; Muñoz-González et al., 2017; Yang et al., 2017). Koh and Liang (2017) called this the projected gradient ascent method since it calculates the gradient during training, but instead of changing the model parameters to decrease the loss, it poisons the data to increase the loss. This attack on accuracy can be defined as the following optimization problem, where $\epsilon$ is a hyperparameter discussed in section 3.4.

$$\max_{\mathcal{D}_p} L_{adv}\left(\hat{\theta}; \mathcal{D}_{\text{test}}\right) \quad \text{s.t. } |\mathcal{D}_p| = \epsilon |\mathcal{D}_c| \quad \text{with} \quad \mathcal{D}_p \subseteq \mathcal{F}_\beta$$
$$\text{where } \hat{\theta} = \arg\min \mathcal{L}\left(\theta; \mathcal{D}_c \cup \mathcal{D}_p\right). \tag{1}$$

**Influence Attack on Fairness** (IAF) is a DPA inspired by the influence attack on accuracy (Koh and Liang, 2017) and the work of Zafar et al. (2015), which introduced a loss function for fair classification involving a fairness constraint, called decision boundary covariance. Decision boundary covariance is the covariance between the sensitive feature $z$, which is gender in this case, and the signed distance from the feature vector to the decision boundary $d_\theta(x)$.

---

[2]https://docs.scipy.org/doc/scipy/reference/generated/scipy.optimize.fmin.html

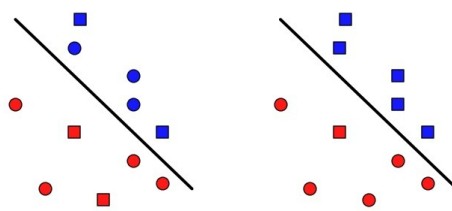

Figure 1: The shape of the data points represents the sensitive attribute, and the color their labels. The decision boundary is represented by the black line.

$$Cov(z, d_\theta(x)) \approx \frac{1}{N} \sum_{i=1}^{N} (z_i - z) d_\theta(x_i) \tag{2}$$

If class labels in the training set are correlated with one or more sensitive attributes $z_i{}_{i=1}^{N}$ (e.g. gender, race), the percentage of samples with a certain sensitive attribute having $d_\theta(x_i) \geq 0$ may differ drastically from the percentage of users without this sensitive attribute value having $d_\theta(x_i) \geq 0$. The intuition behind decision boundary covariance is that the sensitive attributes should not determine which side of the decision boundary a point is on, and thus which label it receives. The left side of Figure 1 shows an instance where the sensitive attribute (shape) and assigned label (color) have zero covariance, indicating that the sensitive attribute has no influence on classification. On the right, the covariance is either extremely positive or extremely negative, indicating that the sensitive attribute does correlate with the classification result.

The goal of the adversary is to maximize the covariance between $z$ and $d_\theta(x_i)$, which will decrease the fairness of the classification. It is worth noting that this covariance can happen even if sensitive attributes aren't utilized to construct the decision boundary, because sensitive attributes can be correlated with one or more of the other features.

IAF is a variant of the influence attack by Koh et al. (2018) and Koh and Liang (2017) that includes demographic information. This demographic information, specifically gender, is used to decide which group is advantaged and disadvantaged, called $D_a$ and $D_d$ respectively, during sampling. Similar to the convention in Koh et al. (2018), one positive and one negative instance are sampled uniformly at random, after which $|D_c|$ instances are created to act as poisoned points $D_p$. The poisoned data points are inversely proportional to the class balance, such that $(|D_c^+|)$ positive poisoned data points are sampled from $D_a$ and $(|D_c^+|)$ negative poisoned data points are sampled from $D_d$, in which $|D_c^+|$ and $|D_c^-|$ represent the number of positive and negative points in the clean data respectively.

The loss function of IAF, combines $\ell_{fairness}$ with the loss function of the influence attack, $\ell_{acc}$ as defined in Equation 3, with hyperparameter $\lambda$ controlling the impact of the fairness loss on the adversarial loss.

$$L_{adv}(\theta; D_{test}) = \ell_{acc} + \lambda \ell_{fairness} \qquad \text{where} \qquad l_{fairness} = \frac{1}{N} \sum_{i=1}^{N} (z_i - z) d_\theta(x_i) \tag{3}$$

Algorithm 1, as shown in Figure 11, details the implementation of this poisoning attack, using the aforementioned parameters.

**Anchoring Attack** is another DPA and its objective is to target some points and cloud their labels with poisoned points with opposing labels, resulting in a skewed decision boundary. In contrast to IAF, the loss of the model is not used, meaning this attack can be used in combination with any model and loss function.

A target point $x_{target}$ is sampled in one of two ways, as demonstrated in Figure 11. In the random anchoring attack (RAA), these anchor points are chosen uniformly at random for each demographic group, while in the non-random instance (NRAA) they are picked based on their popularity, which is defined as the amount of similar data points in their vicinity. Next, poisoned points are created and are placed in close vicinity of $x_{target}$, resulting in them having the same demographic as $x_{target}$, but the opposite label. This will skew the decision boundary, causing more advantaged

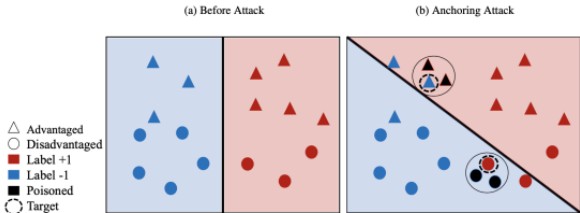

Figure 2: The left figure show a data set before attack and the right figure is an anchoring attack representation displaying how poisoned points are placed in close vicinity (depicted by the large solid circle) to the target points (Mehrabi et al., 2020)

.

points to have a predictive outcome of +1 and more disadvantaged points to have a predictive outcome of -1, as depicted in Figure 2.

## 3.2 Data sets

The data sets listed below were used in both the original paper's experiment and our own. The data is split 80-20 between the training and test set and and gender has been chosen as the sensitive attribute for each data set.

**German Credit data set**[3] This data set is from the UCI ML repository (Dua and Graff, 2017). It contains credit profiles with 20 attributes for 1000 individuals. The classification goal is to predict whether an individual has a good or bad credit score. The pre-processed German data provided with the paper has the same number of samples, but 58 attributes instead of 20. Based on data exploration we made the assumption that this is the result of one-hot-encoding of categorical features.

**COMPAS data set**[4] This data set is provided by ProPublica (Larson et al., 2016). It consists of profiles with 52 attributes such as criminal history, jail time and demographics about 7214 defendants from Broward County. In this case the classification goal is to predict whether an individual will re-offend within two years after being released[5]. The original paper only looked at the eight attributes specified in Table 1. The pre-processed COMPAS data provided with the paper has the same number of samples as the original but 16 attributes, again due to one-hot-encoding.

**Drug Consumption data set**[6] This data set is also from the UCI ML repository (Dua and Graff, 2017). It contains profiles of 1885 individuals, consisting of 32 attributes. The classification goal is to predict whether or not an individual has consumed cocaine at some point in their lifetime. Only the 13 attributes specified in Table 1 are used in the original experiments and our own. The pre-processed drug data provided with the code had 1885 samples and 13 attributes, like the original.

| COMPAS | | Drug | | | |
|---|---|---|---|---|---|
| sex | age_cat | ID | Age | Gender | SS |
| juv_fel_count | juv_misd_count | Education | Country | Ethnicity | |
| priors_count | c_charge_degree | Nscore | Escore | Oscore | |
| race | juv_other_count | Ascore | Cscore | Impulsive | |

Table 1: The features used for the COMPAS and Drug data set

The authors provide pre-processed versions of the aforementioned data sets without a description of the pre-processing methodology. As such, we made the decision to pre-process the raw data we obtained from the original sources and will further refer to these as the recreated data sets.

---

[3] https://archive.ics.uci.edu/ml/datasets/statlog+(german+credit+data)
[4] https://www.propublica.org/datastore/dataset/compas-recidivism-risk-score-data-and-analysis
[5] Therefore, COMPAS-scores-two-years.csv is the relevant data set
[6] https://archive.ics.uci.edu/ml/datasets/Drug+consumption+%28quantified%29

Our pre-processing procedure is based on data exploration of the data sets provided by the authors. To run the code, the sensitive feature index must be specified. Data exploration revealed that the sensitive feature indexes are 36, 4 and 12 respectively for German, COMPAS and Drug. For the sake of simplicity, the sensitive feature is always moved to index 0 in the recreated data sets. Furthermore, males were represented with 0 and females with 1, as this is how they were labeled in the code. After finding out that the number of attributes in the recreated data set did not match the number of attributes of the authors' data, it was discovered that one-hot encoding was used for categorical features, which could explain the reason these data sets contained more attributes than indicated in their paper. Thereafter the data was standardized, with a mean of 0 and a standard deviation of 1. To see whether the attribute values matched the attribute values of the authors' data, all the attributes were compared and popped once they matched. This procedure revealed the index of the sensitive features as well. Finally, the data was shuffled as this is common practice.

### 3.3 Extension

Our contribution to the existing work is making the original paper more reproducible, by documenting how we reproduced the findings for their novel fairness attacks. This is done by providing the pre-processing procedure of the data[7], which was discussed in section 3.2. Furthermore, we organized the code by removing unnecessary code and adding some documentation. This paper also covers all the assumptions made and information obtained from the code that was used to reproduce the results, shown in section 4. This is accumulated into a more comprehensive model description in section 3.1 and experimental setup in section 3.4.

### 3.4 Experimental setup and Computational requirements

**The hyperparameters** for this experiment are $\epsilon$ and $\lambda$. $\epsilon$ determines the size of the poisoned data set as a fraction of the clean data and $\lambda$ controls the trade-off between accuracy loss and fairness loss, in the loss function of IAF; $L_{adv} = \ell_{acc} + \lambda\ell_{fairness}$.

**Statistical Parity Difference** captures the difference in predictive outcome between different (advantaged and disadvantaged) demographic groups. It is defined as:

$$SPD = |p(\hat{Y} = +1|x \in D_a) - p(\hat{Y} = +1|x \in D_d)| \tag{4}$$

**Equality of Opportunity Difference** captures the difference in the true positive rate between different (advantaged and disadvantaged) demographic groups. It is defined as:

$$EOD = |p(\hat{Y} = +1|x \in D_a, Y = +1) - p(\hat{Y} = +1|x \in D_d, Y = +1)| \tag{5}$$

As in the original paper, we evaluate the attacks by plotting accuracy and the aforementioned SPD and EOD fairness criteria. The model becomes more unfair as SPD and EOD get closer to 1. Despite the fact that the authors do not indicate the seed used in their experiment or if they averaged numerous seeds, the code revealed a default seed for each attack setup. In our experiment, three runs were executed for each type of fairness attack and data set. The used seeds for each attack and data set combination were the default seed, the default seed plus 1 and the default seed plus 2. Each run examined $\epsilon$ values ranging from 0.0 to 1.0, with 0.1 increments. $\lambda$ was set to 1.0 for all runs with IAF, like in the original work. Because the original results are only presented as graphs, instead of numbers, we examine the difference between the original and reproduced plots to assess if the reproduced results are similar to the results in the original paper.

It was not specified whether the average accuracy, max accuracy or the last iteration's accuracy was taken over multiple runs. We plotted the results for each instance - an example is given in Figure 6 in the Appendix - and observed that the last of the metrics, accuracy, SPD and EOD is most similar to the results in the original paper. Therefore, metrics of the last iteration are used in Section 4.

Furthermore, the code is not optimised to utilize a GPU, so the experiments are executed on a MacBook Pro (2017) with a 3.3 GHz Dual-Core Intel Core i5 processor and 16 GB memory. The training time was about five minutes for IFA with 30 to 200 iterations, less than one minute for RAA with 29 iterations and less than two minutes for NRAA with 29 iterations. However, the training time for NRAA on the COMPAS data set was about 90 minutes. See Table 3 in the Appendix for further specifics regarding run times.

---

[7]https://anonymous.4open.science/r/Fairness-C81D

 # 4   Results

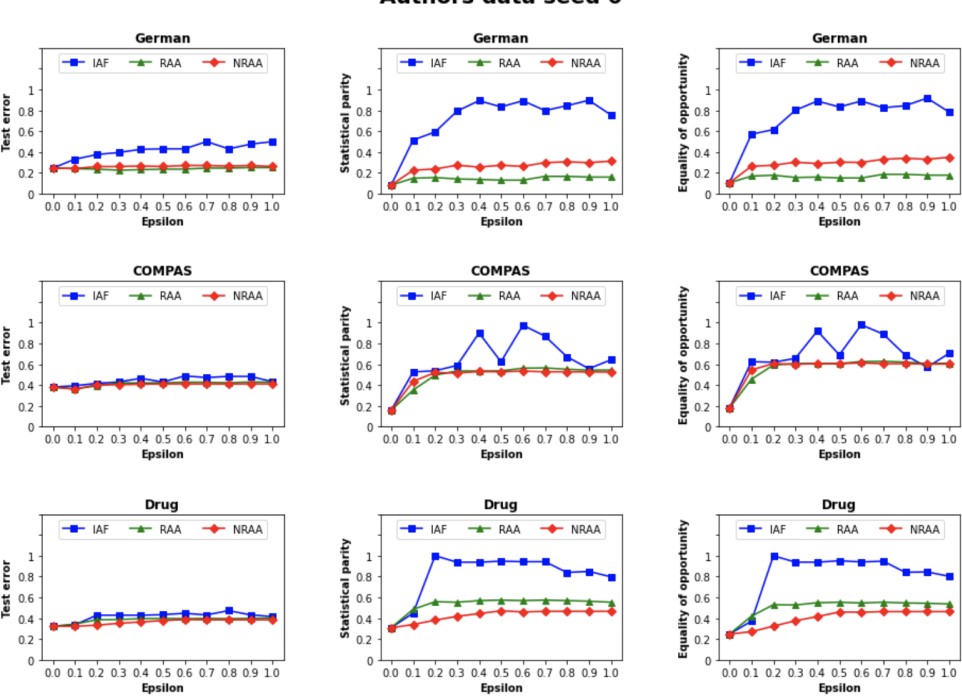

Figure 3: Results obtained for the novel fairness attacks using the default seed and data sets provided by Mehrabi et al. (2020)

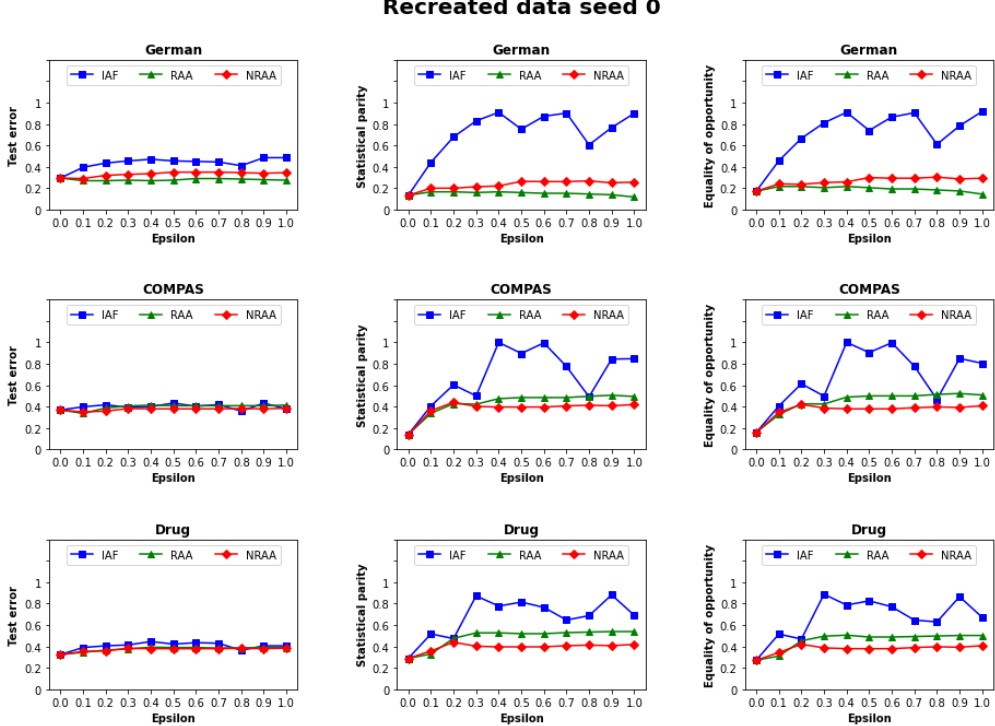

Figure 4: Results obtained for the novel fairness attacks using the default seed and the recreated data sets

### 4.1 Results reproducing the original paper

The results in Figure 3 display the last iteration's accuracy, SPD and EOD, obtained using the data provided with the default seed. The influence attack and both anchoring attacks are presented in the same plot. The reproduced results are similar to those presented by the authors, see Figure 5 in the Appendix. Because the SPD and EOD scores are relatively high for IAF, RAA and NRAA, the results support both claims 1 and 2 from Section 2.

### 4.2 Results beyond original paper

The results in Figure 4 display the last iteration's accuracy, SPD and EOD of the recreated data sets, with the default seed. Although the results differ from the results obtained when using the data provided by the authors, the SPD and EOD scores are relatively high for IAF, RAA and NRAA and therefore, these results also support claim 1 and 2 in section 2. Furthermore, the results for the last iteration's accuracy, SPD and EOD with different seeds for both the authors' data as well as the recreated data are shown in figures 7, 8, 9 and 10.

## 5 Discussion

Upon visual inspection, the results obtained using the authors' data sets, seen in Figure 3, are similar to those presented in their paper, with the graphs following similar patterns as those in the original paper. Small differences may be caused by our assumption that the default seed was used and not an average over various seeds. The results obtained from the recreated data sets, seen in Figure 4, do not appear very similar to those in the original paper. This could be the result of any of the assumptions that needed to be made to recreate the authors' altered data sets, such as the assumption that the data had been shuffled. If any of our assumptions are incorrect, this could well explain the differences. They do, however, follow a similar pattern. It can thus be stated that claims 1 and 2 of the authors are supported by our experimental results.

**Future work** could be to test the robustness of fairness methods using the novel fairness attacks. This was beyond the scope of the work done in Mehrabi et al. (2020), but would be a sensible next step to take, as they were designed for this purpose. Another way in which this work can be expanded upon is by thoroughly comparing these results to those of attacks on accuracy to test claim 3 as listed in Section 2. Also, these results can be compared with the results of other fairness attacks to better contextualize the performances of the novel attacks. Additionally, it can be of interest to test the fairness performance of the novel attacks on different data sets with sensitive attributes other than gender to see how well the attacks generalize.

### 5.1 What was easy and what was difficult

Once the digital environment was received from the authors, we were able to run the code with the provided data sets and obtain results similar to those given in the original paper, see Figure 4.1.

However, the lack of documentation in the method regarding the type of model used, the data pre-processing procedure, a lack of details regarding SVM and hinge loss make the original paper unnecessarily time-consuming to reproduce. A significant amount of the information about the implementation, needed to reproduce the experiments from scratch, was provided by the code they released and their reference materials, such as Koh et al. (2018) and Zafar et al. (2015).

### 5.2 Communication with original authors

There was no direct communication between us and the original authors. However, we communicated with other research teams working on reproducing the same work and they provided us with a digital environment file supplied by the authors that is not publicly available. Its content is listed in the Appendix.

## 6 Conclusion

It can be concluded that the main claims of Mehrabi et al. (2020) regarding the effectiveness of their fairness attacks are correct. However, fully reproducing their results proved too difficult with our setup. The main obstacles we encountered were a lack of documentation regarding their data pre-processing and their used model. Future work would do well to focus on several areas, such as comparisons with other attacks or experimentation with different data sets.

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

## Appendix

### List of used dependencies

- Python 3.6
- PIP 20.3.1
- setuptools 19.2 (in most of the cases you have to downgrade)
- Tensorflow 1.12.3
- scikit-learn 0.23.1
- tensorboard 1.12.2
- cvxpy 0.4.11 [cvxpy 1.0+ is not backwards compatible, therefore the downgrade of setuptools]
- CVXcanon 0.1.1
- scs 2.1.2
- scipy 1.1.0
- numpy 1.16.2
- pandas 1.1.4
- Matplotlib 3.3.3
- tabulate 0.8.9
- seaborn 0.11.0
- tqdm 4.62.3
- IPython 7.16.1
- pillow 8.0.1

### List of Assumptions Made

- The seed used by the authors is the default seed observed in the code.
- Data was shuffled before use
- Categorical features were one-hot encoded except the sensitive feature.
- Female is represented with the value 1 and male with the value 0.
- Data was standardized with a mean of 0 and a standard deviation of 1
- Results were based on the test error, SPD and EOD of the last iteration.
- The feasible set is assumed to be decided by simply projecting the data to a sphere or slab within the vicinity of the target

| Abbreviation | Meaning | Page |
|---|---|---|
| ML | Machine learning | 2 |
| SPD | Statistical parity difference | 2 |
| EOD | Equality of oppurtunity difference | 2 |
| IAF | Influence attack on fairness | 2 |
| DPA | Data poisoning attack | 3 |
| RAA | Random anchoring attack | 3 |
| NRAA | Non-random anchoring attack | 3 |

Table 2: Summary of Abbreviations

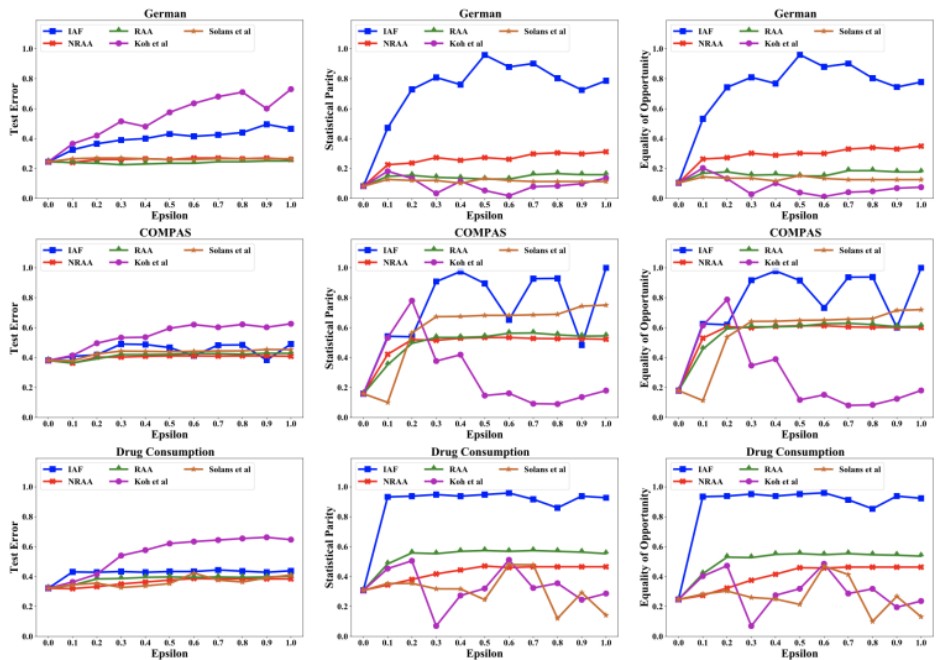

Figure 5: Results obtained for different attacks (Mehrabi et al., 2020)

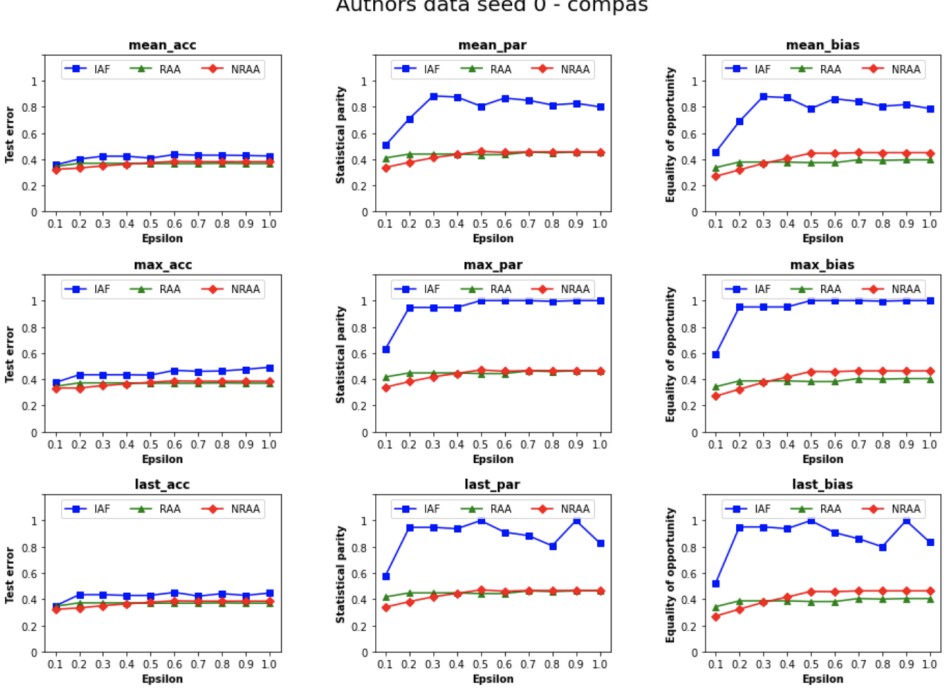

Figure 6: Results obtained for different attacks with different metrics: mean, max and last.

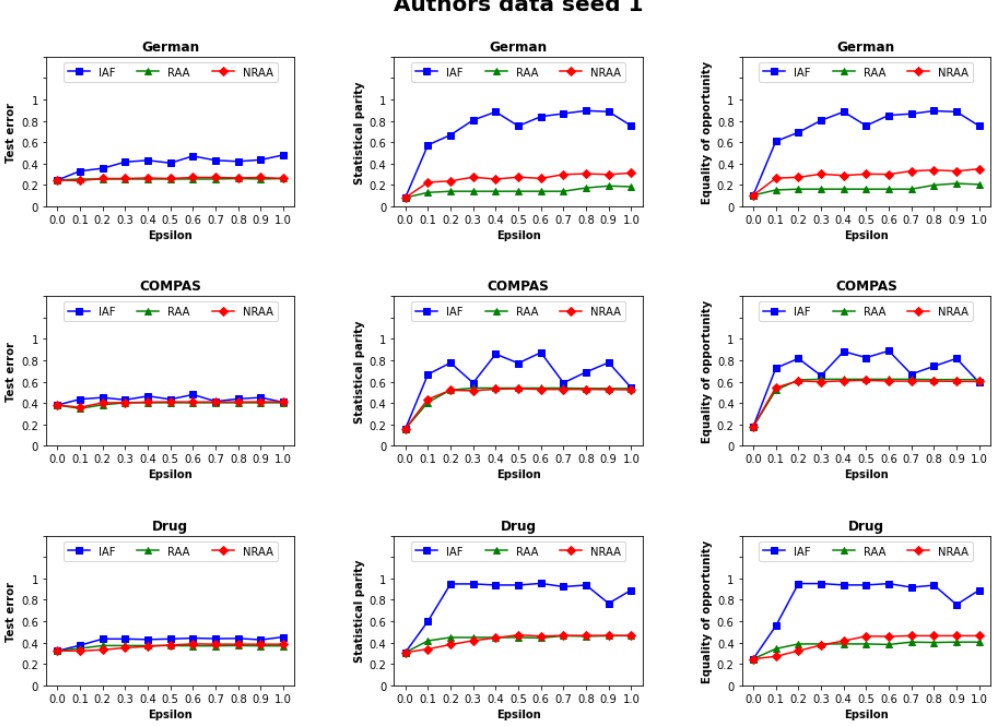

Figure 7: Results obtained for different attacks using seed 1 and data sets provided by Mehrabi et al. (2020)

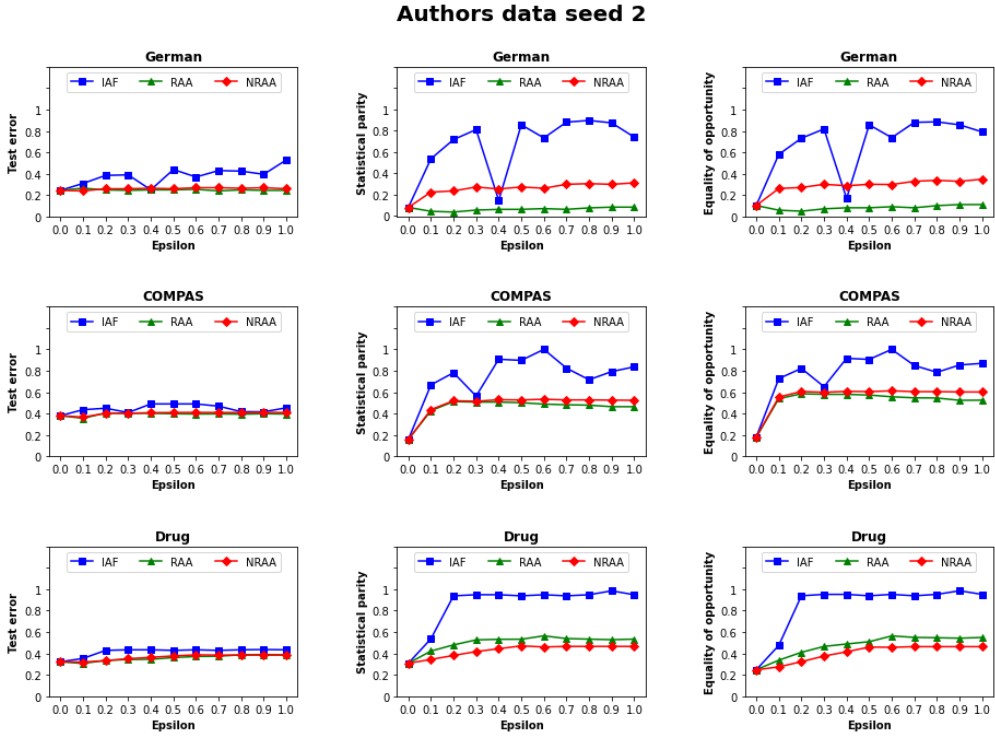

Figure 8: Results obtained for different attacks using seed 2 and data sets provided by Mehrabi et al. (2020)

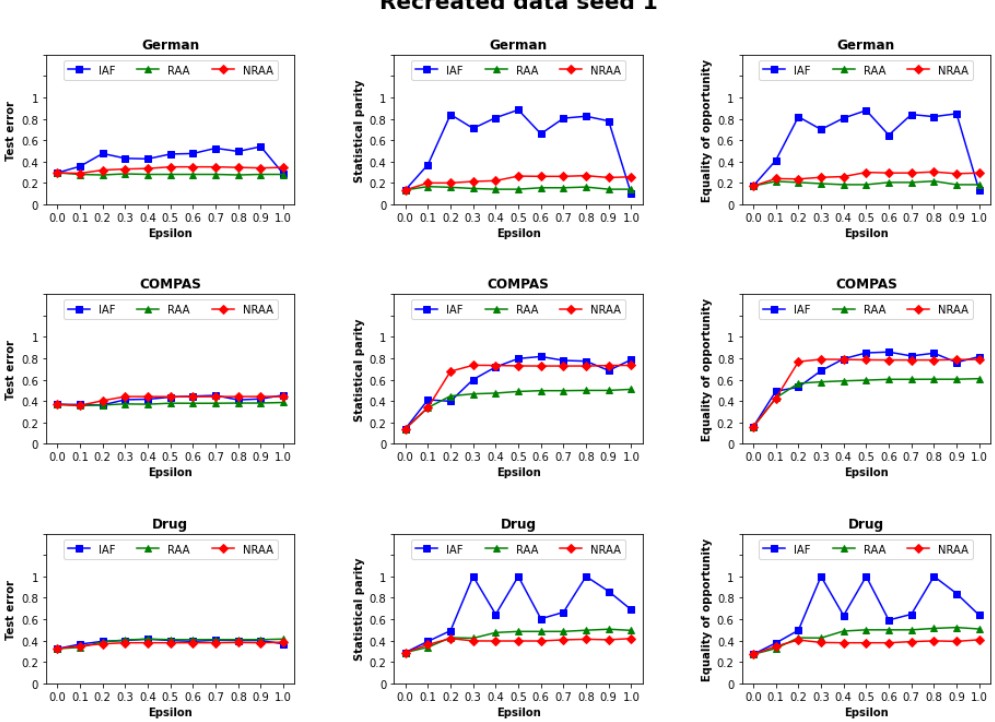

Figure 9: Results obtained for the novel fairness attacks using seed 1 and the recreated data sets

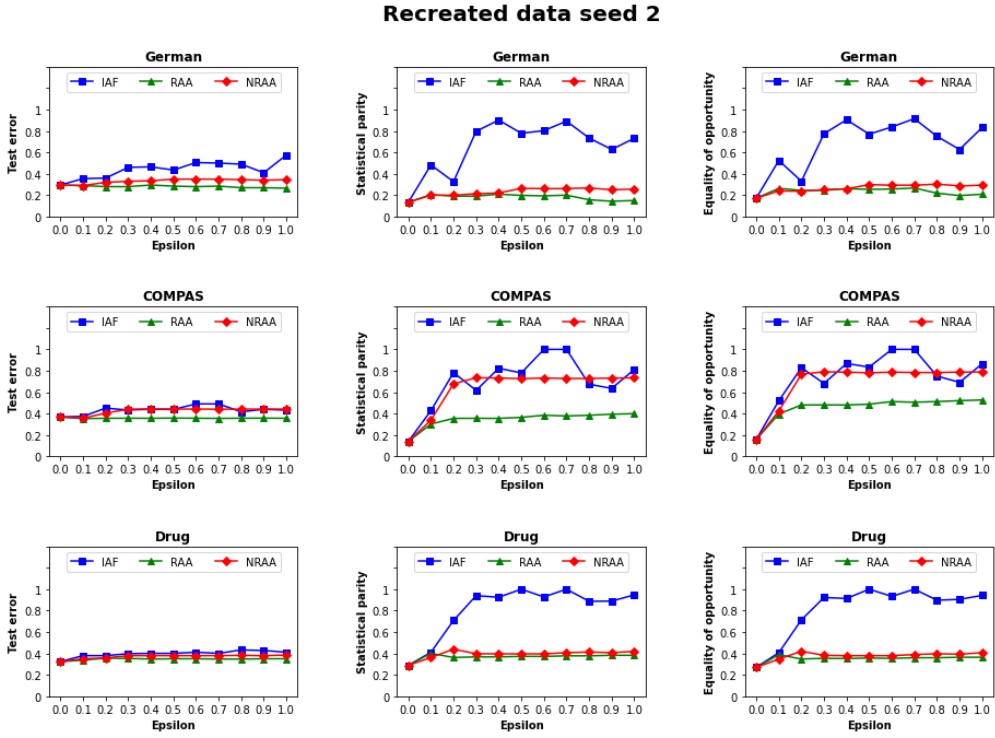

Figure 10: Results obtained for the novel fairness attacks using seed 2 and the recreated data sets

**Algorithm 1:** Influence Attack on Fairness

Input: clean data set
  $\mathcal{D}_c = \{(x_1, y_1), (x_2, y_2), ..., (x_n, y_n)\}$, poison
  fraction $\epsilon$, and step size $\eta$.
Output: poisoned data set
  $\mathcal{D}_p = \{(\tilde{x}_1, \tilde{y}_1), (\tilde{x}_2, \tilde{y}_2), ..., (\tilde{x}_{\epsilon n}, \tilde{y}_{\epsilon n})\}$.
From $\mathcal{D}_a$ randomly sample the positive poisoned
  instance $\mathcal{I}_+ \leftarrow (\tilde{x}_1, \tilde{y}_1)$.
From $\mathcal{D}_d$ randomly sample the negative poisoned
  instance $\mathcal{I}_- \leftarrow (\tilde{x}_2, \tilde{y}_2)$.
Make copies from $\mathcal{I}_+$ and $\mathcal{I}_-$ until having $\epsilon|\mathcal{D}_c|$
  poisoned copies $\mathcal{C}_p$.
Load poisoned data set $\mathcal{D}_p \leftarrow \{\mathcal{C}_p\}$.
Load feasible set by applying anomaly detector $B$
  $\mathcal{F}_\beta \leftarrow B(\mathcal{D}_c \cup \mathcal{D}_p)$.
**for** $t= 1,2,...$ **do**
  $\hat{\theta} \leftarrow argmin_\theta \mathcal{L}(\theta; (\mathcal{D}_c \cup \mathcal{D}_p))$.
  Pre-compute $g_{\hat{\theta}, \mathcal{D}_{test}}^\top H_{\hat{\theta}}^{-1}$ from $L_{adv}$ for details
    refer to (Koh, Steinhardt, and Liang 2018).
  **for** $i= 1,2$ **do**
    Set $\tilde{x}_i^0 \leftarrow \tilde{x}_i - \eta g_{\hat{\theta}, \mathcal{D}_{test}}^\top H_{\hat{\theta}}^{-1} \frac{\partial^2 \ell(\hat{\theta}; \tilde{x}_i, \tilde{y}_i|}{\partial \hat{\theta} \partial \tilde{x}_i}$.
    Set $\tilde{x}_i \leftarrow argmin_{x \in \mathcal{F}_\beta}||x - \tilde{x}_i^0||_2$.   (To
      project $\mathcal{D}_p$ back to $\mathcal{F}_\beta$).
  **end**
  Update copies $\mathcal{C}_p$ based on updates on $\mathcal{I}_+$ and $\mathcal{I}_-$.
  Update feasible set $\mathcal{F}_\beta \leftarrow B(\mathcal{D}_c \cup \mathcal{D}_p)$.
**end**

**Algorithm 2:** Anchoring Attack

Input: clean data set
  $\mathcal{D}_c = \{(x_1, y_1), (x_2, y_2), ..., (x_n, y_n)\}$, poison
  fraction $\epsilon$, and vicinity distance $\tau$.
Output: poisoned data set
  $\mathcal{D}_p = \{(\tilde{x}_1, \tilde{y}_1), (\tilde{x}_2, \tilde{y}_2), ..., (\tilde{x}_{\epsilon n}, \tilde{y}_{\epsilon n})\}$.
**for** $t= 1,2,...$ **do**
  Sample negative $x_{target-}$ from $\mathcal{D}_a$ and positive
    $x_{target+}$ from $\mathcal{D}_d$ with random or non-random
    technique.
  $\mathcal{G}_+$: Generate $(|\mathcal{D}_c^-|\epsilon)$ positive poisoned points
    $(\tilde{x}_+, +1)$ with $\mathcal{D}_a$ in the close vicinity of
    $x_{target-}$ s.t. $||\tilde{x}_+ - x_{target-}||_2 \leq \tau$.
  $\mathcal{G}_-$: Generate $(|\mathcal{D}_c^+|\epsilon)$ negative poisoned points
    $(\tilde{x}_-, -1)$ with $\mathcal{D}_d$ in the close vicinity of
    $x_{target+}$ s.t. $||\tilde{x}_- - x_{target+}||_2 \leq \tau$.
  Load $\mathcal{D}_p$ from the generated data above
    $\mathcal{D}_p \leftarrow \mathcal{G}_+ \cup \mathcal{G}_-$.
  Load the feasible set $\mathcal{F}_\beta \leftarrow B(\mathcal{D}_c \cup \mathcal{D}_p)$.
  **for** $i=1,2,...,\epsilon n$ **do**
    Set $\tilde{x}_i \leftarrow argmin_{x \in \mathcal{F}_\beta}||x - \tilde{x}_i||_2$.   (To
      project $\mathcal{D}_p$ back to $\mathcal{F}_\beta$).
  **end**
  $argmin_\theta \mathcal{L}(\theta; (\mathcal{D}_c \cup \mathcal{D}_p))$.
**end**

Figure 11: Left: IAF algorithm. Right: Anchoring attack algorithm, as described in Mehrabi et al. (2020)

|  | IAF | | RAA | | NRAA | |
| --- | --- | --- | --- | --- | --- | --- |
|  | Time (s) | # iters | Time (s) | # iters | Time (s) | # iters |
| COMPAS | 88.0 | 77.0 | 33.0 | 28.0 | 5411.0 | 28.0 |
| Drug consumption | 50.0 | 67.0 | 20.0 | 28.0 | 170.0 | 28.0 |
| German | 203.0 | 143.0 | 30.0 | 28.0 | 67.0 | 28.0 |

Table 3: Summary of time and iterations needed to run each data set

