# OpenReview forum: "Reproducibility Study of ’Exacerbating Algorithmic Bias through Fairness Attacks’"
_ML_Reproducibility_Challenge/2021/Fall — RC2021_

### Official Review · Reviewer_Jqgy · 2022-02-23
**Review for “Reproducibility Study of ’Exacerbating Algorithmic Bias through Fairness  Attacks’”**

**Rating:** 5
**Confidence:** 3

**Review:**

Scope of reproducibility

The report presents clearly the scope of reproducibility and adheres to it.

Code

The code of the original author is re-used. The code is submitted with an extension of how
do the authors perform the pre-processing the datasets.

Communication with original authors

There is no direct contact between the authors and the original authors.

Hyperparameter Search
The authors use the hyperparameters as described in the original paper.

Discussion on results

The reproduction results are discussed clearly. The easy and difficult parts in the
reproducing are clearly presented.

Recommendations for reproducibility

The authors mention that recreating the datasets (to obtain similar results) requires several
assumptions.

Results beyond the paper

There is no extra results beyond the original paper.

Overall organization and clarity

The writing is good.

---

### Official Review · Reviewer_Uqzz · 2022-03-01
**Good summary of preprocessing used in study and reproduction of results**

**Rating:** 7
**Confidence:** 4

**Review:**

This paper provides a decent summary of pre-processing procedure that is missing in the original paper. It establishes empirically that the last iteration metrics were reported in the original paper. The results support the claims in the original paper albeit the results are slightly different. They gave an explanation why the results might be different but it is not entirely clear that it is the only reason.

It is good that authors also tried different seeds and provided results. I think it is a good paper and will help reimplement original paper easier.

Even though the paper is easy to read there are small typos:
In line 109, I think it is not clear that x refers to feature vector.
In equation 2 and 3 and other places where its used on RHS, it should be average of z but it is not written.
Lines 125 and 126, need to check the variable $D_c^+$ and $D_c^-$
Might make line 199 better and clear

---

### Official Review · Reviewer_Mny6 · 2022-03-19
**Review: Reproducibility Study of ’Exacerbating Algorithmic Bias through Fairness Attacks’**

**Rating:** 8
**Confidence:** 5

**Review:**

- Reproducibility Summary: Present
- Scope of reproducibility: Clearly Stated; adheres to it
- Code: re-used author repository
- Communication with original authors: yes; though in-direct communication (via other teams working on the same paper)
- Hyperparameter Search: yes; includes new hyperparameters not tried by the authors
- Ablation Study: Comprehensive
- Discussion on results: Extensive discussion; detailed description of the reproducible/non-reproducible parts
- Recommendations for reproducibility: Useful criticism for the authors
- Results beyond the paper: Yes;
- Overall organization and clarity: Good

---

### Meta-Review · Area_Chair_NTZW · 2022-04-08

**Recommendation:** Accept
**Confidence:** 4

**Metareview:**

The authors did a great job at reproducing the results of the original paper. Moreover, the overall presentation is good and the results are communicated in a very clear and concise way.

---

### Decision · Program_Chairs · 2022-04-09

**Decision:**

Accept

**Comment:**

Following the recommendation of reviewers and meta-reviewer, the paper is accepted for ML Reproducibility Challenge 2021, and will be published in the upcoming special edition of ReScience Journal.